# INVARIANT AND EQUIVARIANT GRAPH NETWORKS

**Haggai Maron, Heli Ben-Hamu, Nadav Shamir & Yaron Lipman**
Department of Computer Science and Applied Mathematics
Weizmann Institute of Science
Rehovot, Israel

## ABSTRACT

Invariant and equivariant networks have been successfully used for learning images, sets, point clouds, and graphs. A basic challenge in developing such networks is finding the maximal collection of invariant and equivariant *linear* layers. Although this question is answered for the first three examples (for popular transformations, at-least), a full characterization of invariant and equivariant linear layers for graphs is not known.

In this paper we provide a characterization of all permutation invariant and equivariant linear layers for (hyper-)graph data, and show that their dimension, in case of edge-value graph data, is 2 and 15, respectively. More generally, for graph data defined on $k$-tuples of nodes, the dimension is the $k$-th and $2k$-th Bell numbers. Orthogonal bases for the layers are computed, including generalization to multigraph data. The constant number of basis elements and their characteristics allow successfully applying the networks to different size graphs. From the theoretical point of view, our results generalize and unify recent advancement in equivariant deep learning. In particular, we show that our model is capable of approximating any message passing neural network.

Applying these new linear layers in a simple deep neural network framework is shown to achieve comparable results to state-of-the-art and to have better expressivity than previous invariant and equivariant bases.

## 1 INTRODUCTION

We consider the problem of graph learning, namely finding a functional relation between input graphs (more generally, hyper-graphs) $\mathcal{G}^\ell$ and corresponding targets $T^\ell$, e.g., labels. As graphs are common data representations, this task received quite a bit of recent attention in the machine learning community Bruna et al. (2013); Henaff et al. (2015); Monti et al. (2017); Ying et al. (2018).

More specifically, a (hyper-)graph data point $\mathcal{G} = (\mathbb{V}, \mathbf{A})$ consists of a set of $n$ nodes $\mathbb{V}$, and values $\mathbf{A}$ attached to its *hyper-edges*[1]. These values are encoded in a tensor $\mathbf{A}$. The order of the tensor $\mathbf{A}$, or equivalently, the number of indices used to represent its elements, indicates the type of data it represents, as follows: First order tensor represents *node-values* where $\mathbf{A}_i$ is the value of the $i$-th node; Second order tensor represents *edge-values*, where $\mathbf{A}_{ij}$ is the value attached to the $(i,j)$ edge; in general, $k$-th order tensor encodes *hyper-edge-values*, where $\mathbf{A}_{i_1,\dots,i_k}$ represents the value of the hyper-edge represented by $(i_1, \dots, i_k)$. For example, it is customary to represent a graph using a binary adjacency matrix $\mathbf{A}$, where $\mathbf{A}_{ij}$ equals one if vertex $i$ is connected to vertex $j$ and zero otherwise. We denote the set of order-$k$ tensors by $\mathbb{R}^{n^k}$.

The task at hand is constructing a functional relation $f(\mathbf{A}^\ell) \approx T^\ell$, where $f$ is a neural network. If $T^\ell = t^\ell$ is a single output response then it is natural to ask that $f$ is *order invariant*, namely it should produce the same output regardless of the node numbering used to encode $\mathbf{A}$. For example, if we represent a graph using an adjacency matrix $\mathbf{A} = A \in \mathbb{R}^{n \times n}$, then for an arbitrary permutation matrix $\boldsymbol{P}$ and an arbitrary adjacency matrix $\boldsymbol{A}$, the function $f$ is order invariant if it satisfies $f(\boldsymbol{P}^T \boldsymbol{A} \boldsymbol{P}) = f(\boldsymbol{A})$. If the targets $T^\ell$ specify output response in a form of a tensor, $T^\ell = \mathbf{T}^\ell$, then it is natural to ask that $f$ is *order equivariant*, that is, $f$ commutes with the renumbering of nodes operator acting on tensors. Using the above adjacency matrix example, for every adjacency matrix $\boldsymbol{A}$ and

---

[1]A hyper-edge is an ordered subset of the nodes, $\mathbb{V}$

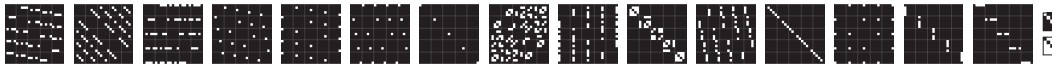

Figure 1: The full basis for equivariant linear layers for edge-value data $\mathbf{A} \in \mathbb{R}^{n \times n}$, for $n = 5$. The purely linear 15 basis elements, $\mathbf{B}^\mu$, are represented by matrices $n^2 \times n^2$, and the 2 bias basis elements (right), $\mathbf{C}^\lambda$, by matrices $n \times n$, see equation 9.

every permutation matrix $\boldsymbol{P}$, the function $f$ is equivariant if it satisfies $f(\boldsymbol{P}^T \boldsymbol{A} \boldsymbol{P}) = \boldsymbol{P}^T f(\boldsymbol{A}) \boldsymbol{P}$. To define invariance and equivariance for functions acting on general tensors $\mathbf{A} \in \mathbb{R}^{n^k}$ we use the *reordering operator*: $\boldsymbol{P} \star \mathbf{A}$ is defined to be the tensor that results from renumbering the nodes $\mathbb{V}$ according to the permutation defined by $\boldsymbol{P}$. Invariance now reads as $f(\boldsymbol{P} \star \mathbf{A}) = f(\mathbf{A})$; while equivariance means $f(\boldsymbol{P} \star \mathbf{A}) = \boldsymbol{P} \star f(\mathbf{A})$. Note that the latter equivariance definition also holds for functions between different order tensors, $f : \mathbb{R}^{n^k} \to \mathbb{R}^{n^l}$.

Following the standard paradigm of neural-networks where a network $f$ is defined by alternating compositions of linear layers and non-linear activations, we set as a goal to characterize all *linear* invariant and equivariant layers. The case of node-value input $\mathbf{A} = \boldsymbol{a} \in \mathbb{R}^n$ was treated in the pioneering works of Zaheer et al. (2017); Qi et al. (2017). These works characterize all linear permutation invariant and equivariant operators acting on node-value (i.e., first order) tensors, $\mathbb{R}^n$. In particular it it shown that the linear space of invariant linear operators $L : \mathbb{R}^n \to \mathbb{R}$ is of dimension one, containing essentially only the sum operator, $L(\boldsymbol{a}) = \alpha \boldsymbol{1}^T \boldsymbol{a}$. The space of equivariant linear operators $L : \mathbb{R}^n \to \mathbb{R}^n$ is of dimension two, $L(\boldsymbol{a}) = \left[ \alpha \boldsymbol{I} + \beta(\boldsymbol{1}\boldsymbol{1}^T - \boldsymbol{I}) \right] \boldsymbol{a}$.

The general equivariant tensor case was partially treated in Kondor et al. (2018) where the authors make the observation that the set of standard tensor operators: product, element-wise product, summation, and contraction are all equivariant, and due to linearity the same applies to their linear combinations. However, these do not exhaust nor provide a full and complete basis for *all* possible tensor equivariant linear layers.

In this paper we provide a full characterization of permutation invariant and equivariant linear layers for general tensor input and output data. We show that the space of invariant linear layers $L : \mathbb{R}^{n^k} \to \mathbb{R}$ is of dimension $\mathrm{b}(k)$, where $\mathrm{b}(k)$ is the $k$-th *Bell number*. The $k$-th Bell number is the number of possible partitions of a set of size $k$; see inset for the case $k = 3$. Furthermore, the space of equivariant linear layers 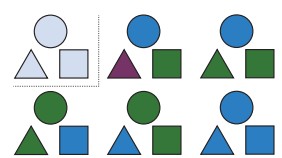

$L : \mathbb{R}^{n^k} \to \mathbb{R}^{n^l}$ is of dimension $\mathrm{b}(k + l)$. Remarkably, this dimension is independent of the size $n$ of the node set $\mathbb{V}$. This allows applying the same network on graphs of different sizes. For both types of layers we provide a general formula for an orthogonal basis that can be readily used to build linear invariant or equivariant layers with maximal expressive power. Going back to the example of a graph represented by an adjacency matrix $\boldsymbol{A} \in \mathbb{R}^{n \times n}$ we have $k = 2$ and the linear invariant layers $L : \mathbb{R}^{n \times n} \to \mathbb{R}$ have dimension $\mathrm{b}(2) = 2$, while linear equivariant layers $L : \mathbb{R}^{n \times n} \to \mathbb{R}^{n \times n}$ have dimension $\mathrm{b}(4) = 15$. Figure 1 shows visualization of the basis to the linear equivariant layers acting on edge-value data such as adjacency matrices.

In Hartford et al. (2018) the authors provide an impressive generalization of the case of node-value data to several node sets, $\mathbb{V}_1, \mathbb{V}_2, \ldots, \mathbb{V}_m$ of sizes $n_1, n_2, \ldots, n_m$. Their goal is to learn interactions across sets. That is, an input data point is a tensor $\mathbf{A} \in \mathbb{R}^{n_1 \times n_2 \times \cdots \times n_m}$ that assigns a value to each element in the cartesian product $\mathbb{V}_1 \times \mathbb{V}_2 \times \cdots \times \mathbb{V}_m$. Renumbering the nodes in each node set using permutation matrices $\boldsymbol{P}_1, \ldots, \boldsymbol{P}_m$ (resp.) results in a new tensor we denote by $\boldsymbol{P}_{1:m} \star \mathbf{A}$. Order invariance means $f(\boldsymbol{P}_{1:m} \star \mathbf{A}) = f(\mathbf{A})$ and order equivariance is $f(\boldsymbol{P}_{1:m} \star \mathbf{A}) = \boldsymbol{P}_{1:m} \star f(\mathbf{A})$. Hartford et al. (2018) introduce bases for linear invariant and equivariant layers. Although the layers in Hartford et al. (2018) satisfy the order invariance and equivariance, they do not exhaust all possible such layers in case some node sets coincide. For example, if $\mathbb{V}_1 = \mathbb{V}_2$ they have 4 independent learnable parameters where our model has the maximal number of 15 parameters.

Our analysis allows generalizing the multi-node set case to arbitrary tensor data over $\mathbb{V}_1 \times \mathbb{V}_2 \times \cdots \times \mathbb{V}_m$. Namely, for data points in the form of a tensor $\mathbf{A} \in \mathbb{R}^{n_1^{k_1} \times n_2^{k_2} \times \cdots \times n_m^{k_m}}$. The tensor $\mathbf{A}$ attaches a value to every element of the Cartesian product $\mathbb{V}_1^{k_1} \times \cdots \times \mathbb{V}_2^{k_2}$, that is, $k_1$-tuple from $\mathbb{V}_1$, $k_2$-tuple from $\mathbb{V}_2$ and so forth. We show that the linear space of invariant linear layers $L : \mathbb{R}^{n_1^{k_1} \times n_2^{k_2} \times \cdots \times n_m^{k_m}} \to \mathbb{R}$ is of dimension $\prod_{i=1}^m \mathrm{b}(k_i)$, while the equivariant linear layers $L :$

$\mathbb{R}^{n_1^{k_1} \times n_2^{k_2} \times \cdots \times n_m^{k_m}} \rightarrow \mathbb{R}^{n_1^{l_1} \times n_2^{l_2} \times \cdots \times n_m^{l_m}}$ has dimension $\prod_{i=1}^{m} \mathrm{b}(k_i + l_i)$. We also provide orthogonal bases for these spaces. Note that, for clarity, the discussion above disregards biases and features; we detail these in the paper.

In appendix C we show that our model is capable of approximating any message-passing neural network as defined in Gilmer et al. (2017) which encapsulate several popular graph learning models. One immediate corollary is that the universal approximation power of our model is not lower than message passing neural nets.

In the experimental part of the paper we concentrated on possibly the most popular instantiation of graph learning, namely that of a single node set and edge-value data, e.g., with adjacency matrices. We created simple networks by composing our invariant or equivariant linear layers in standard ways and tested the networks in learning invariant and equivariant graph functions: (i) We compared identical networks with our basis and the basis of Hartford et al. (2018) and showed we can learn graph functions like trace, diagonal, and maximal singular vector. The basis in Hartford et al. (2018), tailored to the multi-set setting, cannot learn these functions demonstrating it is not maximal in the graph-learning (i.e., multi-set with repetitions) scenario. We also demonstrate our representation allows extrapolation: learning on one size graphs and testing on another size; (ii) We also tested our networks on a collection of graph learning datasets, achieving results that are comparable to the state-of-the-art in 3 social network datasets.

## 2  PREVIOUS WORK

Our work builds on two main sub-fields of deep learning: group invariant or equivariant networks, and deep learning on graphs. Here we briefly review the relevant works.

**Invariance and equivariance in deep learning.**    In many learning tasks the functions that we want to learn are invariant or equivariant to certain symmetries of the input object description. Maybe the first example is the celebrated *translation invariance* of Convolutional Neural Networks (CNNs) (LeCun et al., 1989; Krizhevsky et al., 2012); in this case, the image label is invariant to a translation of the input image. In recent years this idea was generalized to other types of symmetries such as rotational symmetries (Cohen & Welling, 2016a;b; Weiler et al., 2018; Cohen et al., 2018). Cohen & Welling (2016a) introduced Group Equivariant Neural Networks that use a generalization of the convolution operator to groups of rotations and reflections; Weiler et al. (2018); Cohen et al. (2018) also considered rotational symmetries but in the case of 3D shapes and spherical functions. Ravanbakhsh et al. (2017) showed that any equivariant layer is equivalent to a certain parameter sharing scheme. If we adopt this point of view, our work reveals the structure of the parameter sharing in the case of graphs and hyper-graphs. In another work, Kondor & Trivedi (2018) show that a neural network layer is equivariant to the action of some compact group iff it implements a generalized form of the convolution operator. Yarotsky (2018) suggested certain group invariant/equivariant models and proved their universality. To the best of our knowledge these models were not implemented.

**Learning of graphs.**    Learning of graphs is of huge interest in machine learning and we restrict our attention to recent advancements in deep learning on graphs. Gori et al. (2005); Scarselli et al. (2009) introduced Graph Neural Networks (GNN): GNNs hold a state (a real valued vector) for each node in the graph, and propagate these states according to the graph structure and learned parametric functions. This idea was further developed in Li et al. (2015) that use gated recurrent units. Following the success of CNNs, numerous works suggested ways to define convolution operator on graphs. One promising approach is to define convolution by imitating its spectral properties using the Laplacian operator to define generalized Fourier basis on graphs (Bruna et al., 2013). Multiple follow-up works (Henaff et al., 2015; Defferrard et al., 2016; Kipf & Welling, 2016; Levie et al., 2017) suggest more efficient and spatially localized filters. The main drawback of spectral approaches is that the generalized Fourier basis is graph-dependent and applying the same network to different graphs can be challenging. Another popular way to generalize the convolution operator to graphs is learning stationary functions that operate on neighbors of each node and update its current state (Atwood & Towsley, 2016; Duvenaud et al., 2015; Hamilton et al., 2017; Niepert et al., 2016; Veličković et al., 2017; Monti et al., 2017; Simonovsky & Komodakis, 2017). This idea generalizes the *locality* and *weight sharing* properties of the standard convolution operators on regular grids. As shown in the important work of Gilmer et al. (2017), most of the the above mentioned methods (including the spectral methods) can be seen as instances of the general class of *Message Passing Neural Networks*.

## 3 LINEAR INVARIANT AND EQUIVARIANT LAYERS

In this section we characterize the collection of linear invariant and equivariant layers. We start with the case of a single node set $\mathbb{V}$ of size $n$ and edge-value data, that is order 2 tensors $\mathbf{A} = A \in \mathbb{R}^{n \times n}$. As a typical example imagine, as above, an adjacency matrix of a graph. We set a bit of notation. Given a matrix $\boldsymbol{X} \in \mathbb{R}^{a \times b}$ we denote $\text{vec}(\boldsymbol{X}) \in \mathbb{R}^{ab \times 1}$ its column stack, and by brackets the inverse action of reshaping to a square matrix, namely $[\text{vec}(\boldsymbol{X})] = \boldsymbol{X}$. Let $p$ denote an arbitrary permutation and $\boldsymbol{P}$ its corresponding permutation matrix.

Let $\boldsymbol{L} \in \mathbb{R}^{1 \times n^2}$ denote the matrix representing a general linear operator $L : \mathbb{R}^{n \times n} \to \mathbb{R}$ in the standard basis, then $L$ is order invariant iff $\boldsymbol{L}\text{vec}(\boldsymbol{P}^T \boldsymbol{A} \boldsymbol{P}) = \boldsymbol{L}\text{vec}(\boldsymbol{A})$. Using the property of the Kronecker product that $\text{vec}(\boldsymbol{X} \boldsymbol{A} \boldsymbol{Y}) = \boldsymbol{Y}^T \otimes \boldsymbol{X}\text{vec}(\boldsymbol{A})$, we get the equivalent equality $\boldsymbol{L}\boldsymbol{P}^T \otimes \boldsymbol{P}^T\text{vec}(\boldsymbol{A}) = \boldsymbol{L}\text{vec}(\boldsymbol{A})$. Since the latter equality should hold for every $\boldsymbol{A}$ we get (after transposing both sides of the equation) that order invariant $\boldsymbol{L}$ is equivalent to the equation
$$\boldsymbol{P} \otimes \boldsymbol{P}\,\text{vec}(\boldsymbol{L}) = \text{vec}(\boldsymbol{L}) \tag{1}$$
for every permutation matrix $\boldsymbol{P}$. Note that we used $\boldsymbol{L}^T = \text{vec}(\boldsymbol{L})$.

For equivariant layers we consider a general linear operator $L : \mathbb{R}^{n \times n} \to \mathbb{R}^{n \times n}$ and its corresponding matrix $\boldsymbol{L} \in \mathbb{R}^{n^2 \times n^2}$. Equivariance of $L$ is now equivalent to $[\boldsymbol{L}\text{vec}(\boldsymbol{P}^T \boldsymbol{A} \boldsymbol{P})] = \boldsymbol{P}^T[\boldsymbol{L}\text{vec}(\boldsymbol{A})]\boldsymbol{P}$. Using the above property of the Kronecker product again we get $\boldsymbol{L}\boldsymbol{P}^T \otimes \boldsymbol{P}^T\text{vec}(\boldsymbol{A}) = \boldsymbol{P}^T \otimes \boldsymbol{P}^T \boldsymbol{L}\text{vec}(\boldsymbol{A})$. Noting that $\boldsymbol{P}^T \otimes \boldsymbol{P}^T$ is an $n^2 \times n^2$ permutation matrix and its inverse is $\boldsymbol{P} \otimes \boldsymbol{P}$ we get to the equivalent equality $\boldsymbol{P} \otimes \boldsymbol{P}\boldsymbol{L}\boldsymbol{P}^T \otimes \boldsymbol{P}^T\text{vec}(\boldsymbol{A}) = \boldsymbol{L}\text{vec}(\boldsymbol{A})$. As before, since this holds for every $\boldsymbol{A}$ and using the properties of the Kronecker product we get that $\boldsymbol{L}$ is order equivariant iff for all permutation matrices $\boldsymbol{P}$
$$\boldsymbol{P} \otimes \boldsymbol{P} \otimes \boldsymbol{P} \otimes \boldsymbol{P}\,\text{vec}(\boldsymbol{L}) = \text{vec}(\boldsymbol{L}). \tag{2}$$

From equations 1 and 2 we see that finding invariant and equivariant linear layers for the order-2 tensor data over one node set requires finding fixed points of the permutation matrix group represented by Kronecker powers $\boldsymbol{P} \otimes \boldsymbol{P} \otimes \cdots \otimes \boldsymbol{P}$ of permutation matrices $\boldsymbol{P}$. As we show next, this is also the general case for order-$k$ tensor data $\mathbf{A} \in \mathbb{R}^{n^k}$ over one node set, $\mathbb{V}$. That is,
$$\text{invariant } \boldsymbol{L} : \qquad \boldsymbol{P}^{\otimes k}\text{vec}(\boldsymbol{L}) = \text{vec}(\boldsymbol{L}) \tag{3}$$
$$\text{equivariant } \boldsymbol{L} : \qquad \boldsymbol{P}^{\otimes 2k}\text{vec}(\boldsymbol{L}) = \text{vec}(\boldsymbol{L}) \tag{4}$$

for every permutation matrix $\boldsymbol{P}$, where $\boldsymbol{P}^{\otimes k} = \overbrace{\boldsymbol{P} \otimes \cdots \otimes \boldsymbol{P}}^{k}$. In equation 3, $\boldsymbol{L} \in \mathbb{R}^{1 \times n^k}$ is the matrix of an invariant operator; and in equation 4, $\boldsymbol{L} \in \mathbb{R}^{n^k \times n^k}$ is the matrix of an equivariant operator. We call equations 3,4 the *fixed-point equations*.

To see this, let us add a bit of notation first. Let $p$ denote the permutation corresponding to the permutation matrix $\boldsymbol{P}$. We let $\boldsymbol{P} \star \mathbf{A}$ denote the tensor that results from expressing the tensor $\mathbf{A}$ after renumbering the nodes in $\mathbb{V}$ according to permutation $\boldsymbol{P}$. Explicitly, the $(p(i_1), p(i_2), \ldots, p(i_k))$-th entry of $\boldsymbol{P} \star \mathbf{A}$ equals the $(i_1, i_2, \ldots, i_k)$-th entry of $\mathbf{A}$. The matrix that corresponds to the operator $\boldsymbol{P} \star$ in the standard tensor basis $\boldsymbol{e}^{(i_1)} \otimes \cdots \otimes \boldsymbol{e}^{(i_k)}$ is the Kronecker power $\boldsymbol{P}^{T \otimes k} = (\boldsymbol{P}^T)^{\otimes k}$. Note that $\text{vec}(\mathbf{A})$ is exactly the coordinate vector of the tensor $\mathbf{A}$ in this standard basis and therefore we have $\text{vec}(\boldsymbol{P} \star \mathbf{A}) = \boldsymbol{P}^{T \otimes k}\text{vec}(\mathbf{A})$. We now show:

**Proposition 1.** *A linear layer is invariant (equivariant) if and only if its coefficient matrix satisfies the fixed-point equations, namely equation 3 (equation 4).*

*Proof.* Similarly to the argument from the order-2 case, let $\boldsymbol{L} \in \mathbb{R}^{1 \times n^k}$ denote the matrix corresponding to a general linear operator $L : \mathbb{R}^{n^k} \to \mathbb{R}$. Order invariance means
$$\boldsymbol{L}\text{vec}(\boldsymbol{P} \star \mathbf{A}) = \boldsymbol{L}\text{vec}(\mathbf{A}). \tag{5}$$
Using the matrix $\boldsymbol{P}^{T \otimes k}$ we have equivalently $\boldsymbol{L}\boldsymbol{P}^{T \otimes k}\text{vec}(\mathbf{A}) = \boldsymbol{L}\text{vec}(\mathbf{A})$ which is in turn equivalent to $\boldsymbol{P}^{\otimes k}\text{vec}(\boldsymbol{L}) = \text{vec}(\boldsymbol{L})$ for all permutation matrices $\boldsymbol{P}$. For order equivariance, let $\boldsymbol{L} \in \mathbb{R}^{n^k \times n^k}$ denote the matrix of a general linear operator $L : \mathbb{R}^{n^k} \to \mathbb{R}^{n^k}$. Now equivariance of $L$ is equivalent to
$$[\boldsymbol{L}\text{vec}(\boldsymbol{P} \star \mathbf{A})] = \boldsymbol{P} \star [\boldsymbol{L}\text{vec}(\mathbf{A})]. \tag{6}$$
Similarly to above this is equivalent to $\boldsymbol{L}\boldsymbol{P}^{T \otimes k}\text{vec}(\mathbf{A}) = \boldsymbol{P}^{T \otimes k}\boldsymbol{L}\text{vec}(\mathbf{A})$ which in turn leads to $\boldsymbol{P}^{\otimes k}\boldsymbol{L}\boldsymbol{P}^{T \otimes k} = \boldsymbol{L}$, and using the Kronecker product properties we get $\boldsymbol{P}^{\otimes 2k}\text{vec}(\boldsymbol{L}) = \text{vec}(\boldsymbol{L})$. $\square$

### 3.1 SOLVING THE FIXED-POINT EQUATIONS

We have reduced the problem of finding all invariant and equivariant linear operators $L$ to finding all solutions $\boldsymbol{L}$ of equations 3 and 4. Although the fixed point equations consist of an exponential number of equations with only a polynomial number of unknowns they actually possess a solution space of constant dimension (i.e., independent of $n$).

To find the solution of $\boldsymbol{P}^{\otimes \ell}\text{vec}(\mathbf{X}) = \text{vec}(\mathbf{X})$, where $\mathbf{X} \in \mathbb{R}^{n^{\ell}}$, note that $\boldsymbol{P}^{\otimes \ell}\text{vec}(\mathbf{X}) = \text{vec}(\boldsymbol{Q} \star \mathbf{X})$, where $\boldsymbol{Q} = \boldsymbol{P}^T$. As above, the tensor $\boldsymbol{Q} \star \mathbf{X}$ is the tensor resulted from renumbering the nodes in $\mathbb{V}$ using permutation $\boldsymbol{Q}$. Equivalently, the fixed-point equations we need to solve can be formulated as

$$Q \star \mathbf{X} = \mathbf{X}, \quad \forall \boldsymbol{Q} \text{ permutation matrices} \tag{7}$$

The permutation group is acting on tensors $\mathbf{X} \in \mathbb{R}^{n^{\ell}}$ with the action $\mathbf{X} \mapsto \boldsymbol{Q} \star \mathbf{X}$. We are looking for fixed points under this action. To that end, let us define an equivalence relation in the index space of tensors $\mathbb{R}^{n^{\ell}}$, namely in $[n]^{\ell}$, where with a slight abuse of notation (we use light brackets) we set $[n] = \{1, 2, \ldots, n\}$. For multi-indices $\boldsymbol{a}, \boldsymbol{b} \in [n]^{\ell}$ we set $\boldsymbol{a} \sim \boldsymbol{b}$ iff $\boldsymbol{a}, \boldsymbol{b}$ have the same *equality pattern*, that is $\boldsymbol{a}_i = \boldsymbol{a}_j \Leftrightarrow \boldsymbol{b}_i = \boldsymbol{b}_j$ for all $i, j \in [\ell]$.

The equality pattern equivalence relation partitions the index set $[n]^{\ell}$ into equivalence classes, the collection of which is denoted $[n]^{\ell}/_{\sim}$. Each equivalence class can be represented by a unique partition of the set $[\ell]$ where each set in the partition indicates maximal set of identical values. Let us exemplify. For $\ell = 2$ we have two equivalence classes $\gamma_1 = \{\{1\}, \{2\}\}$ and $\gamma_2 = \{\{1, 2\}\}$; $\gamma_1$ represents all multi-indices $(i, j)$ where $i \neq j$, while $\gamma_2$ represents all multi-indices $(i, j)$ where $i = j$. For $\ell = 4$, there are 15 equivalence classes $\gamma_1 = \{\{1\}, \{2\}, \{3\}, \{4\}\}$, $\gamma_2 = \{\{1\}, \{2\}, \{3, 4\}\}$, $\gamma_3 = \{\{1, 2\}, \{3\}, \{4\}\}$, ...; $\gamma_3$ represents multi-indices $(i_1, i_2, i_3, i_4)$ so that $i_1 = i_2$, $i_2 \neq i_3$, $i_3 \neq i_4$, $i_2 \neq i_4$.

For each equivalence class $\gamma \in [n]^{\ell}/_{\sim}$ we define an order-$\ell$ tensor $\mathbf{B}^{\gamma} \in \mathbb{R}^{n^{\ell}}$ by setting

$$\mathbf{B}_{\boldsymbol{a}}^{\gamma} = \begin{cases} 1 & \boldsymbol{a} \in \gamma \\ 0 & \text{otherwise} \end{cases} \tag{8}$$

Since we have a tensor $\mathbf{B}^{\gamma}$ for every equivalence class $\gamma$, and the equivalence classes are in one-to-one correspondence with partitions of the set $[\ell]$ we have $\mathrm{b}(\ell)$ tensors $\mathbf{B}^{\gamma}$. (Remember that $\mathrm{b}(\ell)$ denotes the $\ell$-th Bell number.) We next prove:

**Proposition 2.** *The tensors $\mathbf{B}^{\gamma}$ in equation 8 form an orthogonal basis (in the standard inner-product) to the solution set of equations 7. The dimension of the solution set is therefore $\mathrm{b}(\ell)$.*

*Proof.* Let us first show that: $\mathbf{X}$ is a solution to equation 7 iff $\mathbf{X}$ is constant on equivalence classes of the equality pattern relation, $\sim$. Since permutation $q : [n] \to [n]$ is a bijection the equality patterns of $\boldsymbol{a} = (i_1, i_2, \ldots, i_{\ell}) \in [n]^{\ell}$ and $q(\boldsymbol{a}) = (q(i_1), q(i_2), \ldots, q(i_{\ell})) \in [n]^{\ell}$ are identical, i.e., $\boldsymbol{a} \sim q(\boldsymbol{a})$. Taking the $\boldsymbol{a} \in [n]^{\ell}$ entry of both sides of equation 7 gives $\mathbf{X}_{q(\boldsymbol{a})} = \mathbf{X}_{\boldsymbol{a}}$. Now, if $\mathbf{X}$ is constant on equivalence classes then in particular it will have the same value at $\boldsymbol{a}$ and $q(\boldsymbol{a})$ for all $\boldsymbol{a} \in [n]^{\ell}$ and permutations $q$. Therefore $\mathbf{X}$ is a solution to equation 7. For the only if part, consider a tensor $\mathbf{X}$ for which there exist multi-indices $\boldsymbol{a} \sim \boldsymbol{b}$ (with identical equality patterns) and $\mathbf{X}_{\boldsymbol{a}} \neq \mathbf{X}_{\boldsymbol{b}}$ then $\mathbf{X}$ is not a solution to equation 7. Indeed, since $\boldsymbol{a} \sim \boldsymbol{b}$ one can find a permutation $q$ so that $\boldsymbol{b} = q(\boldsymbol{a})$ and using the equation above, $\mathbf{X}_{\boldsymbol{b}} = \mathbf{X}_{q(\boldsymbol{a})} = \mathbf{X}_{\boldsymbol{a}}$ which leads to a contradiction.
To finish the proof note that any tensor $\mathbf{X}$, constant on equivalence classes, can be written as a linear combination of $\mathbf{B}^{\gamma}$, which are merely indicators of the equivalence class. Furthermore, the collection $\mathbf{B}^{\gamma}$ have pairwise disjoint supports and therefore are an orthogonal basis. $\square$

Combining propositions 1 and 2 we get the characterization of invariant and equivariant linear layers acting on general $k$-order tensor data over a single node set $\mathbb{V}$:

**Theorem 1.** *The space of invariant (equivariant) linear layers $\mathbb{R}^{n^k} \to \mathbb{R}$ ($\mathbb{R}^{n^k} \to \mathbb{R}^{n^k}$) is of dimension $\mathrm{b}(k)$ ($\mathrm{b}(2k)$) with basis elements $\mathbf{B}^{\gamma}$ defined in equation 8, where $\gamma$ are equivalence classes in $[n]^k/_{\sim}$ ($[n]^{2k}/_{\sim}$).*

**Biases** Theorem 1 deals with purely linear layers, that is without *bias*, i.e., without constant part. Nevertheless extending the previous analysis to constant layers is straight-forward. First,

any constant layer $\mathbb{R}^{n^k} \to \mathbb{R}$ is also invariant so all constant invariant layers are represented by constants $c \in \mathbb{R}$. For equivariant layers $L : \mathbb{R}^{n^k} \to \mathbb{R}^{n^k}$ we note that equivariance means $\mathbf{C} = L(\boldsymbol{P} \star \mathbf{A}) = \boldsymbol{P} \star L(\mathbf{A}) = \boldsymbol{P} \star \mathbf{C}$. Representing this equation in matrix form we get $\boldsymbol{P}^{T \otimes k}\text{vec}(\mathbf{C}) = \text{vec}(\mathbf{C})$. This shows that constant equivariant layers on one node set acting on general $k$-order tensors are also characterized by the fixed-point equations, and in fact have the same form and dimensionality as invariant layers on $k$-order tensors, see equation 3. Specifically, their basis is $\mathbf{B}^\lambda$, $\lambda \in [n]^k /_\sim$. For example, for $k = 2$, the biases are shown on the right in figure 1.

**Features.** It is pretty common that input tensors have vector values (i.e., features) attached to each hyper-edge ($k$-tuple of nodes) in $\mathbb{V}$, that is $\mathbf{A} \in \mathbb{R}^{n^k \times d}$. Now linear invariant $\mathbb{R}^{n^k \times d} \to \mathbb{R}^{1 \times d'}$ or equivariant $\mathbb{R}^{n^k \times d} \to \mathbb{R}^{n^k \times d'}$ layers can be formulated using a slight generalization of the previous analysis. The operator $\boldsymbol{P} \star \mathbf{A}$ is defined to act only on the nodal indices, i.e., $i_1, \ldots, i_k$ (the first $k$ indices). Explicitly, the $(p(i_1), p(i_2), \ldots, p(i_k), i_{k+1})$-th entry of $\boldsymbol{P} \star \mathbf{A}$ equals the $(i_1, i_2, \ldots, i_k, i_{k+1})$-th entry of $\mathbf{A}$.

Invariance is now formulated exactly as before, equation 5, namely $\boldsymbol{L}\text{vec}(\boldsymbol{P} \star \mathbf{A}) = \boldsymbol{L}\text{vec}(\mathbf{A})$. The matrix that corresponds to $\boldsymbol{P}\star$ acting on $\mathbb{R}^{n^k \times d}$ in the standard basis is $\boldsymbol{P}^{T \otimes k} \otimes \boldsymbol{I}_d$ and therefore $\boldsymbol{L}(\boldsymbol{P}^{T \otimes k} \otimes \boldsymbol{I}_d)\text{vec}(\mathbf{A}) = \boldsymbol{L}\text{vec}(\mathbf{A})$. Since this is true for all $\mathbf{A}$ we have $(\boldsymbol{P}^{\otimes k} \otimes \boldsymbol{I}_d \otimes \boldsymbol{I}_{d'})\,\text{vec}(\boldsymbol{L}) = \text{vec}(\boldsymbol{L})$, using the properties of the Kronecker product. Equivariance is written as in equation 6, $[\boldsymbol{L}\text{vec}(\boldsymbol{P} \star \mathbf{A})] = \boldsymbol{P} \star [\boldsymbol{L}\text{vec}(\mathbf{A})]$. In matrix form, the equivariance equation becomes $\boldsymbol{L}(\boldsymbol{P}^{T \otimes k} \otimes \boldsymbol{I}_d)\text{vec}(\mathbf{A}) = (\boldsymbol{P}^{T \otimes k} \otimes \boldsymbol{I}_{d'})\boldsymbol{L}\text{vec}(\mathbf{A})$, since this is true for all $\mathbf{A}$ and using the properties of the Kronecker product again we get to $\boldsymbol{P}^{\otimes k} \otimes \boldsymbol{I}_d \otimes \boldsymbol{P}^{\otimes k} \otimes \boldsymbol{I}_{d'}\,\text{vec}(\boldsymbol{L}) = \text{vec}(\boldsymbol{L})$. The basis (with biases) to the solution space of these fixed-point equations is defined as follows. We use $\boldsymbol{a}, \boldsymbol{b} \in [n]^k$, $i, j \in [d]$, $i', j' \in [d']$, $\lambda \in [n]^k /_\sim$, $\mu \in [n]^{2k} /_\sim$.

$$\text{invariant:} \qquad \mathbf{B}^{\lambda,j,j'}_{\boldsymbol{a},i,i'} = \begin{cases} 1 & \boldsymbol{a} \in \lambda,\ i=j,\ i'=j' \\ 0 & \text{otherwise} \end{cases} \quad ; \quad \mathbf{C}^{j'}_{i'} = \begin{cases} 1 & i'=j' \\ 0 & \text{otherwise} \end{cases} \tag{9a}$$

$$\text{equivariant:} \qquad \mathbf{B}^{\mu,j,j'}_{\boldsymbol{a},i,\boldsymbol{b},i'} = \begin{cases} 1 & (\boldsymbol{a},\boldsymbol{b}) \in \mu,\ i=j,\ i'=j' \\ 0 & \text{otherwise} \end{cases} \quad ; \quad \mathbf{C}^{\lambda,j'}_{\boldsymbol{b},i'} = \begin{cases} 1 & \boldsymbol{b} \in \lambda,\ i'=j' \\ 0 & \text{otherwise} \end{cases} \tag{9b}$$

Note that these basis elements are similar to the ones in equation 8 with the difference that we have different basis tensor for each pair of input $j$ and output $j'$ feature channels.

An invariant (equation 10a)/ equivariant (equation 10b) linear layer $L$ including the biases can be written as follows for input $\mathbf{A} \in \mathbb{R}^{n^k \times d}$:

$$L(\mathbf{A})_{i'} = \sum_{\boldsymbol{a},i} \mathbf{T}_{\boldsymbol{a},i,i'} \mathbf{A}_{\boldsymbol{a},i} + \mathbf{Y}_{i'}; \quad \mathbf{T} = \sum_{\lambda,j,j'} w_{\lambda,j,j'} \mathbf{B}^{\lambda,j,j'}; \mathbf{Y} = \sum_{j'} b_{j'} \mathbf{C}^{j'} \tag{10a}$$

$$L(\mathbf{A})_{\boldsymbol{b},i'} = \sum_{\boldsymbol{a},i} \mathbf{T}_{\boldsymbol{a},i,\boldsymbol{b},i'} \mathbf{A}_{\boldsymbol{a},i} + \mathbf{Y}_{\boldsymbol{b},i'}; \quad \mathbf{T} = \sum_{\mu,j,j'} w_{\mu,j,j'} \mathbf{B}^{\mu,j,j'}; \mathbf{Y} = \sum_{\lambda,j'} b_{\lambda,j'} \mathbf{C}^{\lambda,j'} \tag{10b}$$

where the learnable parameters are $w \in \mathbb{R}^{\text{b}(k) \times d \times d'}$ and $b \in \mathbb{R}^{d'}$ for a single linear *invariant* layer $\mathbb{R}^{n^k \times d} \to \mathbb{R}^{d'}$; and it is $w \in \mathbb{R}^{\text{b}(2k) \times d \times d'}$ and $b \in \mathbb{R}^{\text{b}(k) \times d'}$ for a single linear *equivariant* layer $\mathbb{R}^{n^k \times d} \to \mathbb{R}^{n^k \times d'}$. The natural generalization of theorem 1 to include bias and features is therefore:

**Theorem 2.** *The space of invariant (equivariant) linear layers $\mathbb{R}^{n^k,d} \to \mathbb{R}^{d'}$ ($\mathbb{R}^{n^k \times d} \to \mathbb{R}^{n^k \times d'}$) is of dimension $dd'\text{b}(k) + d'$ (for equivariant: $dd'\text{b}(2k) + d'\text{b}(k)$) with basis elements defined in equation 9; equation 10a (10b) show the general form of such layers.*

Since, by similar arguments to proposition 2, the purely linear parts $\mathbf{B}$ and biases $\mathbf{C}$ in equation 9 are independent solutions to the relevant fixed-point equations, theorem 2 will be proved if their number equals the dimension of the solution space of these fixed-point equations, namely $dd'\text{b}(k)$ for purely linear part and $d'$ for bias in the invariant case, and $dd'\text{b}(2k)$ for purely linear and $d'\text{b}(k)$ for bias in the equivariant case. This can be shown by repeating the arguments of the proof of proposition 2 slightly adapted to this case, or by a combinatorial identity we show in Appendix B .

For example, figure 1 depicts the 15 basis elements for linear equivariant layers $\mathbb{R}^{n \times n} \to \mathbb{R}^{n \times n}$ taking as input edge-value (order-2) tensor data $\mathbf{A} \in \mathbb{R}^{n \times n}$ and outputting the same dimension tensor. The basis for the purely linear part are shown as $n^2 \times n^2$ matrices while the bias part as $n \times n$ matrices (far right); the size of the node set is $|\mathbb{V}| = n = 5$.

**Mixed order equivariant layers.** Another useful generalization of order equivariant linear layers is to linear layers between *different order* tensor layers, that is, $L : \mathbb{R}^{n^k} \to \mathbb{R}^{n^l}$, where $l \neq k$. For example, one can think of a layer mapping an adjacency matrix to per-node features. For simplicity we will discuss the purely linear scalar-valued case, however generalization to include bias and/or general feature vectors can be done as discussed above. Consider the matrix $\boldsymbol{L} \in \mathbb{R}^{n^l \times n^k}$ representing the linear layer $L$, using the renumbering operator, $\boldsymbol{P}\star$, order equivariance is equivalent to $[\boldsymbol{L}\mathrm{vec}(\boldsymbol{P} \star \mathsf{A})] = \boldsymbol{P} \star [\boldsymbol{L}\mathrm{vec}(\mathsf{A})]$. Note that while this equation looks identical to equation 6 it is nevertheless different in the sense that the $\boldsymbol{P}\star$ operator in the l.h.s. of this equation acts on $k$-order tensors while the one on the r.h.s. acts on $l$-order tensor. Still, we can transform this equation to a matrix equation as before by remembering that $\boldsymbol{P}^{T\otimes k}$ is the matrix representation of the renumbering operator $\boldsymbol{P}\star$ acting on $k$-tensors in the standard basis. Therefore, repeating the arguments in proof of proposition 1, equivariance is equivalent to $\boldsymbol{P}^{\otimes(k+l)}\mathrm{vec}(\boldsymbol{L}) = \mathrm{vec}(\boldsymbol{L})$, for all permutation matrices $\boldsymbol{P}$. This equation is solved as in section 3.1. The corresponding bases to such equivariant layers are computed as in equation 9b, with the only difference that now $\boldsymbol{a} \in [n]^k$, $\boldsymbol{b} \in [n]^l$, and $\mu \in [n]^{k+l}/_\sim$.

## 4 EXPERIMENTS

**Implementation details.** We implemented our method in Tensorflow (Abadi et al., 2016). The equivariant linear basis was implemented efficiently using basic row/column/diagonal summation operators, see appendix A for details. The networks we used are composition of $1 - 4$ equivariant linear layers with ReLU activation between them for the equivariant function setting. For invariant function setting we further added a max over the invariant basis and $1 - 3$ fully-connected layers with ReLU activations.

Table 1: Comparison to baseline methods on synthetic experiments.

| | Symmetric projection | | | Diagonal extraction | | | Max singular vector | | | | Trace | | |
|---|---|---|---|---|---|---|---|---|---|---|---|---|---|
| # Layers | 1 | 2 | 3 | 1 | 2 | 3 | 1 | 2 | 3 | 4 | 1 | 2 | 3 |
| Trivial predictor | 4.17 | 4.17 | 4.17 | 0.21 | 0.21 | 0.21 | 0.025 | 0.025 | 0.025 | 0.025 | 333.33 | 333.33 | 333.33 |
| Hartford et al. | 2.09 | 2.09 | 2.09 | 0.81 | 0.81 | 0.81 | 0.043 | 0.044 | 0.043 | 0.043 | 316.22 | 311.55 | 307.97 |
| Ours | **1E-05** | **7E-06** | **2E-05** | **8E-06** | **7E-06** | **1E-04** | **0.015** | **0.0084** | **0.0054** | **0.0016** | **0.005** | **0.001** | **0.003** |

**Synthetic datasets.** We tested our method on several synthetic equivariant and invariant graph functions that highlight the differences in expressivity between our linear basis and the basis of Hartford et al. (2018). Given an input matrix data $\boldsymbol{A} \in \mathbb{R}^{n \times n}$ we considered: (i) projection onto the symmetric matrices $\frac{1}{2}(\boldsymbol{A}+\boldsymbol{A}^T)$; (ii) diagonal extraction $\mathrm{diag}(\mathrm{diag}(\boldsymbol{A}))$ (keeps only the diagonal and plugs zeros elsewhere); (iii) computing the maximal right singular vector $\arg\max_{\|\boldsymbol{v}\|_2 = 1} \|\boldsymbol{A}\boldsymbol{v}\|_2$; and (iv) computing the trace $\mathrm{tr}(\boldsymbol{A})$. Tasks (i)-(iii) are equivariant while task (iv) is invariant. We created accordingly 4 datasets with $10K$ train and $1K$ test examples of $40 \times 40$ matrices; for tasks (i), (ii), (iv) we used i.i.d. random matrices with uniform distribution in $[0, 10]$; we used mean-squared error (MSE) as loss; for task (iii) we random matrices with uniform distribution of singular values in $[0, 0.5]$ and spectral gap $\geq 0.5$; due to sign ambiguity in this task we used cosine loss of the form $l(\boldsymbol{x}, \boldsymbol{y}) = 1 - \langle \boldsymbol{x}/\|\boldsymbol{x}\|, \boldsymbol{y}/\|\boldsymbol{y}\|\rangle^2$.

We trained networks with 1, 2, and 3 hidden layers with 8 feature channels each and a single fully-connected layer. Both our models as well as Hartford et al. (2018) use the same architecture but with different bases for the linear layers. Table 1 logs the *best* mean-square error of each method over a set of hyper-parameters. We add the MSE for the trivial mean predictor.

This experiment emphasizes simple cases in which the additional parameters in our model, with respect to Hartford et al. (2018), are needed. We note that Hartford et al. (2018) target a different scenario where the permutations acting on the rows and columns of the input matrix are not necessarily the same. The assumption taken in this paper, namely, that the same permutation acts on both rows and

Table 2: Generalization.

| | 30 | 40 | 50 |
|---|---|---|---|
| sym | 0.0053 | 3.8E-05 | 0.0013 |
| svd | 0.0108 | 0.0084 | 0.0096 |
| diag | 0.0150 | 1.5E-05 | 0.0055 |

columns, gives rise to additional parameters that are associated with the diagonal and with the transpose of the matrix (for a complete list of layers for the $k = 2$ case see appendix A). In case of an input matrix that represents graphs, these parameters can be understood as parameters that control self-edges or node features, and incoming/outgoing edges in a different way. Table 2 shows the result of applying the learned equivariant networks from the above experiment to graphs (matrices) of

unseen sizes of $n = 30$ and $n = 50$. Note, that although the network was trained on a fixed size, the network provides plausible generalization to different size graphs. We note that the generalization of the invariant task of computing the trace did not generalize well to unseen sizes and probably requires training on different sizes as was done in the datasets below.

Table 3: Graph Classification Results.

| dataset | MUTAG | PTC | PROTEINS | NCI1 | NCI109 | COLLAB | IMDB-B | IMDB-M |
|---|---|---|---|---|---|---|---|---|
| size | 188 | 344 | 1113 | 4110 | 4127 | 5000 | 1000 | 1500 |
| classes | 2 | 2 | 2 | 2 | 2 | 3 | 2 | 3 |
| avg node # | 17.9 | 25.5 | 39.1 | 29.8 | 29.6 | 74.4 | 19.7 | 13 |
| Results | | | | | | | | |
| DGCNN | 85.83±1.7 | 58.59±2.5 | 75.54±0.9 | 74.44±0.5 | NA | 73.76±0.5 | 70.03±0.9 | 47.83±0.9 |
| PSCN (k=10) | 88.95±4.4 | 62.29±5.7 | 75±2.5 | 76.34±1.7 | NA | 72.6±2.2 | 71±2.3 | 45.23±2.8 |
| DCNN | NA | NA | 61.29±1.6 | 56.61± 1.0 | NA | 52.11±0.7 | 49.06±1.4 | 33.49±1.4 |
| ECC | 76.11 | NA | NA | 76.82 | 75.03 | NA | NA | NA |
| DGK | 87.44±2.7 | 60.08±2.6 | 75.68±0.5 | 80.31±0.5 | 80.32±0.3 | 73.09±0.3 | 66.96±0.6 | 44.55±0.5 |
| DiffPool | NA | NA | 78.1 | NA | NA | 75.5 | NA | NA |
| CCN | 91.64±7.2 | 70.62±7.0 | NA | 76.27±4.1 | 75.54±3.4 | NA | NA | NA |
| GK | 81.39±1.7 | 55.65±0.5 | 71.39±0.3 | 62.49±0.3 | 62.35±0.3 | NA | NA | NA |
| RW | 79.17±2.1 | 55.91±0.3 | 59.57±0.1 | > 3 days | NA | NA | NA | NA |
| PK | 76±2.7 | 59.5±2.4 | 73.68±0.7 | 82.54±0.5 | NA | NA | NA | NA |
| WL | 84.11±1.9 | 57.97±2.5 | 74.68±0.5 | 84.46±0.5 | 85.12±0.3 | NA | NA | NA |
| FGSD | 92.12 | 62.80 | 73.42 | 79.80 | 78.84 | 80.02 | 73.62 | 52.41 |
| AWE-DD | NA | NA | NA | NA | NA | 73.93±1.9 | 74.45 ±5.8 | 51.54 ± 3.6 |
| AWE-FB | 87.87±9.7 | NA | NA | NA | NA | 70.99 ± 1.4 | 73.13 ±3.2 | 51.58 ± 4.6 |
| ours | 84.61±10 | 59.47±7.3 | 75.19±4.3 | 73.71±2.6 | 72.48±2.5 | 77.92±1.7 | 71.27±4.5 | 48.55±3.9 |

**Graph classification.** We tested our method on standard benchmarks of graph classification. We use 8 different real world datasets from the benchmark of Yanardag & Vishwanathan (2015): five of these datasets originate from bioinformatics while the other three come from social networks. In all datasets the adjacency matrix of each graph is used as input and a categorial label is assigned as output. In the bioinformatics datasets node labels are also provided as inputs. These node labels can be used in our framework by placing their 1-hot representations on the diagonal of the input.

Table 3 specifies the results for our method compared to state-of-the-art deep and non-deep graph learning methods. We follow the evaluation protocol including the 10-fold splits of Zhang et al. (2018). For each dataset we selected learning and decay rates on one random fold. In all experiments we used a fixed simple architecture of 3 layers with $(16, 32, 256)$ features accordingly. The last equivariant layer is followed by an invariant max layer according to the invariant basis. We then add two fully-connected hidden layers with $(512, 256)$ features.
We compared our results to seven deep learning methods: DGCNN (Zhang et al., 2018), PSCN (Niepert et al., 2016), DCNN (Atwood & Towsley, 2016), ECC (Simonovsky & Komodakis, 2017), DGK (Yanardag & Vishwanathan, 2015), DiffPool (Ying et al., 2018) and CCN (Kondor et al., 2018). We also compare our results to four popular graph kernel methods: Graphlet Kernel (GK) (Shervashidze et al., 2009),Random Walk Kernel (RW) (Vishwanathan et al., 2010), Propagation Kernel (PK) (Neumann et al., 2016), and Weisfeiler-lehman kernels (WL) (Shervashidze et al., 2011) and two recent feature-based methods: Family of Graph Spectral Distance (FGSD) (Verma & Zhang, 2017) and Anonymous Walk Embeddings (AWE) (Ivanov & Burnaev, 2018). Our method achieved results comparable to the state-of-the-art on the three social networks datasets, and slightly worse results than state-of-the-art on the biological datasets.

## 5 GENERALIZATIONS TO MULTI-NODE SETS

Lastly, we provide a generalization of our framework to data that is given on tuples of nodes from a *collection* of node sets $\mathbb{V}_1, \mathbb{V}_2, \ldots, \mathbb{V}_m$ of sizes $n_1, n_2, \ldots, n_m$ (resp.), namely $\mathbf{A} \in \mathbb{R}^{n_1^{k_1} \times n_2^{k_2} \times \cdots \times n_m^{k_m}}$. We characterize invariant linear layers $L : \mathbb{R}^{n_1^{k_1} \times \cdots \times n_m^{k_m}} \to \mathbb{R}$ and equivariant linear layer $L : \mathbb{R}^{n_1^{k_1} \times \cdots \times n_m^{k_m}} \to \mathbb{R}^{n_1^{l_1} \times \cdots \times n_m^{l_m}}$, where for simplicity we do not discuss features that can be readily added as discussed in section 3. Note that the case of $k_i = l_i = 1$ for all $i = 1, \ldots, m$ is treated in Hartford et al. (2018). The reordering operator now is built out of permutation matrices $\boldsymbol{P}_i \in \mathbb{R}^{n_i \times n_i}$ ($p_i$ denotes the permutation), $i = 1, \ldots, m$, denoted $\boldsymbol{P}_{1:m}\star$, and defined as follows: the $(p_1(\boldsymbol{a}_1), p_2(\boldsymbol{a}_2), \ldots, p_m(\boldsymbol{a}_m))$-th entry of the tensor $\boldsymbol{P}_{1:m} \star \mathbf{A}$, where $\boldsymbol{a}_i \in [n_i]^{k_i}$ is defined to be the $(\boldsymbol{a}_1, \boldsymbol{a}_2, \ldots, \boldsymbol{a}_m)$-th entry of the tensor $\mathbf{A}$. Rewriting the invariant and equivariant equations, i.e., equation 5, 6, in matrix format, similarly to before, we get the fixed-point equa-

tions: $\boldsymbol{M}\mathrm{vec}(\boldsymbol{L}) = \mathrm{vec}(\boldsymbol{L})$ for invariant, and $\boldsymbol{M} \otimes \boldsymbol{M}\mathrm{vec}(\boldsymbol{L}) = \mathrm{vec}(\boldsymbol{L})$ for equivariant, where $\boldsymbol{M} = \boldsymbol{P}_1^{\otimes k_1} \otimes \cdots \otimes \boldsymbol{P}_m^{\otimes k_m}$. The solution of these equations would be linear combinations of basis tensor similar to equation 9 of the form

$$\text{invariant: } \mathbf{B}_{\boldsymbol{a}_1,\ldots,\boldsymbol{a}_m}^{\lambda_1,\ldots,\lambda_m} = \begin{cases} 1 & \boldsymbol{a}_i \in \lambda_i, \ \forall i \\ 0 & \text{otherwise} \end{cases} ; \quad \text{equivariant: } \mathbf{B}_{\boldsymbol{a}_1,\ldots,\boldsymbol{a}_m,\boldsymbol{b}_1,\ldots,\boldsymbol{b}_m}^{\mu_1,\ldots,\mu_m} = \begin{cases} 1 & (\boldsymbol{a}_i,\boldsymbol{b}_i) \in \mu_i, \ \forall i \\ 0 & \text{otherwise} \end{cases} \quad (11)$$

where $\lambda_i \in [n_i]^{k_i}$, $\mu_i \in [n_i]^{k_i+l_i}$, $\boldsymbol{a} \in [n_i]^{k_i}$, $\boldsymbol{b}_i \in [n_i]^{l_i}$. The number of these tensors is $\prod_{i=1}^{m} \mathrm{b}(i)$ for invariant layers and $\prod_{i=1}^{m} \mathrm{b}(k_i + l_i)$ for equivariant layers. Since these are all linear independent (pairwise disjoint support of non-zero entries) we need to show that their number equal the dimension of the solution of the relevant fixed-point equations above. This can be done again by similar arguments to the proof of proposition 2 or as shown in appendix B. To summarize:

**Theorem 3.** *The linear space of invariant linear layers $L : \mathbb{R}^{n_1^{k_1} \times n_2^{k_2} \times \cdots \times n_m^{k_m}} \to \mathbb{R}$ is of dimension $\prod_{i=1}^{m} \mathrm{b}(k_i)$. The equivariant linear layers $L : \mathbb{R}^{n_1^{k_1} \times n_2^{k_2} \times \cdots \times n_m^{k_m}} \to \mathbb{R}^{n_1^{l_1} \times n_2^{l_2} \times \cdots \times n_m^{l_m}}$ has dimension $\prod_{i=1}^{m} \mathrm{b}(k_i + l_i)$. Orthogonal bases for these layers are listed in equation 11.*

ACKNOWLEDGMENTS

This research was supported in part by the European Research Council (ERC Consolidator Grant, "LiftMatch" 771136) and the Israel Science Foundation (Grant No. 1830/17).

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

## APPENDIX A  EFFICIENT IMPLEMENTATION OF LAYERS

For fast execution of order-2 layers we implemented the following 15 operations which can be easily shown to span the basis discussed in the paper. We denote by $\mathbf{1} \in \mathbb{R}^n$ the vector of all ones.

1. The identity and transpose operations: $L(\boldsymbol{A}) = \boldsymbol{A}$, $L(\boldsymbol{A}) = \boldsymbol{A}^T$.

2. The diag operation: $L(\boldsymbol{A}) = \mathrm{diag}(\mathrm{diag}(\boldsymbol{A}))$.

3. Sum of rows replicated on rows/ columns/ diagonal: $L(\boldsymbol{A}) = \boldsymbol{A}\mathbf{1}\mathbf{1}^T$, $L(\boldsymbol{A}) = \mathbf{1}(\boldsymbol{A}\mathbf{1})^T$, $L(\boldsymbol{A}) = \mathrm{diag}(\boldsymbol{A}\mathbf{1})$.

4. Sum of columns replicated on rows/ columns/ diagonal: $L(\boldsymbol{A}) = \boldsymbol{A}^T\mathbf{1}\mathbf{1}^T$, $L(\boldsymbol{A}) = \mathbf{1}(\boldsymbol{A}^T\mathbf{1})^T$, $L(\boldsymbol{A}) = \mathrm{diag}(\boldsymbol{A}^T\mathbf{1})$.

5. Sum of all elements replicated on all matrix/ diagonal: $L(\boldsymbol{A}) = (\mathbf{1}^T\boldsymbol{A}\mathbf{1}) \cdot \mathbf{1}\mathbf{1}^T$, $L(\boldsymbol{A}) = (\mathbf{1}^T\boldsymbol{A}\mathbf{1}) \cdot \mathrm{diag}(\mathbf{1})$.

6. Sum of diagonal elements replicated on all matrix/diagonal: $L(\boldsymbol{A}) = (\mathbf{1}^T\mathrm{diag}(\boldsymbol{A})) \cdot \mathbf{1}\mathbf{1}^T$, $L(\boldsymbol{A}) = (\mathbf{1}^T\mathrm{diag}(\boldsymbol{A})) \cdot \mathrm{diag}(\mathbf{1})$.

7. Replicate diagonal elements on rows/columns: $L(\boldsymbol{A}) = \mathrm{diag}(\boldsymbol{A})\mathbf{1}^T$, $L(\boldsymbol{A}) = \mathbf{1}\mathrm{diag}(\boldsymbol{A})^T$.

We normalize each operation to have unit max operator norm. We note that in case the input matrix is symmetric, our basis reduces to 11 elements in the first layer. If we further assume the matrix has zero diagonal we get a 6 element basis in the first layer. In both cases our model is more expressive than the 4 element basis of Hartford et al. (2018) and as the output of the first layer (or other inner states) need not be symmetric nor have zero diagonal the deeper layers can potentially make good use of the full 15 element basis.

## APPENDIX B   INVARIANT AND EQUIVARIANT SUBSPACE DIMENSIONS

We prove a useful combinatorial fact as a corollary of proposition 2. This fact will be used later to easily compute the dimensions of more general spaces of invariant and equivariant linear layers. We use the fact that if $V$ is a representation of a finite group $G$ then

$$\phi = \frac{1}{|G|} \sum_{g \in G} g \in \text{End}(V) \tag{12}$$

is a projection onto $V^G = \{\boldsymbol{v} \in V \mid g\boldsymbol{v} = \boldsymbol{v}, \ \forall g \in G\}$, the subspace of fixed points in $V$ under the action of $G$, and consequently that $\text{tr}(\phi) = \dim(V^G)$ (see Fulton & Harris (2013) for simple proofs).

**Proposition 3.** *The following formula holds:*

$$\frac{1}{n!} \sum_{\boldsymbol{P} \in \Pi_n} tr(\boldsymbol{P})^k = \text{b}(k),$$

*where $\Pi_n$ is the matrix permutation group of dimensions $n \times n$.*

*Proof.* In our case, the vector space is the space of order-$k$ tensors and the group acting on it is the matrix group $G = \left\{ \boldsymbol{P}^{\otimes k} \mid \boldsymbol{P} \in \Pi_m \right\}$.

$$\dim(V^G) = \text{tr}(\phi) = \frac{1}{|G|} \sum_{g \in G} \text{tr}(g) = \frac{1}{n!} \sum_{\boldsymbol{P} \in \Pi_n} \text{tr}(\boldsymbol{P}^{\otimes k}) = \frac{1}{n!} \sum_{\boldsymbol{P} \in \Pi_n} \text{tr}(\boldsymbol{P})^k,$$

where we used the multiplicative law of the trace with respect to Kronecker product. Now we use proposition 2 noting that in this case $V^G$ is the solution space of the fixed-point equations. Therefore, $\dim(V^G) = \text{b}(k)$ and the proof is finished. $\qquad\square$

Recall that for a permutation matrix $\boldsymbol{P}$, $\text{tr}(\boldsymbol{P}) = |\{i \in [n] \text{ s.t. } \boldsymbol{P} \text{ fixes } e_i\}|$. Using this, we can interpret the equation in proposition 3 as the $k$-th moment of a random variable counting the number of fixed points of a permutation, with uniform distribution over the permutation group. Proposition 3 proves that the $k$-th moment of this random variable is the $k$-th Bell number.

We can now use proposition 3 to calculate the dimensions of two linear layer spaces: (i) Equivariant layers acting on order-$k$ tensors with features (as in 3); and (ii) multi-node sets (as in section 5).

**Theorem 2.** *The space of invariant (equivariant) linear layers $\mathbb{R}^{n^k, d} \to \mathbb{R}^{d'}$ ($\mathbb{R}^{n^k \times d} \to \mathbb{R}^{n^k \times d'}$) is of dimension $dd'\text{b}(k) + d'$ (for equivariant: $dd'\text{b}(2k) + d'\text{b}(k)$) with basis elements defined in equation 9; equations 10a (10b) show the general form of such layers.*

*Proof.* We prove the dimension formulas for the invariant case. The equivariant case is proved similarly. The solution space for the fixed point equations is the set $V^G$ for the matrix group $G = \left\{ \boldsymbol{P}^{\otimes k} \otimes \boldsymbol{I}_d \otimes \boldsymbol{I}_{d'} \mid \boldsymbol{P} \in \Pi_n \right\}$. Using the projection formula 12 we get that the dimension of the solution subspace, which is the space of invariant linear layers, can be computed as follows:

$$\dim(V^G) = \frac{1}{n!} \sum_{\boldsymbol{P} \in \Pi_n} \text{tr}(\boldsymbol{P})^k \, \text{tr}(I_d)\text{tr}(I_{d'}) = \left( \frac{1}{n!} \sum_{\boldsymbol{P} \in \Pi_n} \text{tr}(\boldsymbol{P})^k \right) \text{tr}(I_d) \, \text{tr}(I_{d'}) = d \cdot d' \cdot \text{b}(k).$$

$\qquad\square$

**Theorem 3.** *The linear space of invariant linear layers $L : \mathbb{R}^{n_1^{k_1} \times n_2^{k_2} \times \cdots \times n_m^{k_m}} \to \mathbb{R}$ is of dimension $\prod_{i=1}^{m} \mathrm{b}(k_i)$. The equivariant linear layers $L : \mathbb{R}^{n_1^{k_1} \times n_2^{k_2} \times \cdots \times n_m^{k_m}} \to \mathbb{R}^{n_1^{l_1} \times n_2^{l_2} \times \cdots \times n_m^{l_m}}$ has dimension $\prod_{i=1}^{m} \mathrm{b}(k_i + l_i)$. Orthogonal bases for these layers are listed in equation 11.*

*Proof.* In this case we get the fixed-point equations: $\boldsymbol{M}\mathrm{vec}(\boldsymbol{L}) = \mathrm{vec}(\boldsymbol{L})$ for invariant, and $\boldsymbol{M} \otimes \boldsymbol{M}\mathrm{vec}(\boldsymbol{L}) = \mathrm{vec}(\boldsymbol{L})$ for equivariant, where $\boldsymbol{M} = \boldsymbol{P}_1^{\otimes k_1} \otimes \cdots \otimes \boldsymbol{P}_m^{\otimes k_m}$. Similarly to the previous theorem, plugging $M$ into equation 12, using the trace multiplication rule and proposition 3 we get the above formulas. $\square$

## APPENDIX C    IMPLEMENTING MESSAGE PASSING WITH OUR MODEL

In this appendix we show that our model can approximate message passing layers as defined in Gilmer et al. (2017) to an arbitrary precision, and consequently that our model is able to approximate any network consisting of several such layers. The key idea is to mimic multiplication of features by the adjacency matrix, which allows summing over local neighborhoods. This can be implemented using our basis.

**Theorem 4.** *Our model can represent message passing layers to an arbitrary precision on compact sets.*

*Proof.* Consider input vertex data $\boldsymbol{H} = (h_u) \in \mathbb{R}^{n \times d}$ ($n$ is the number of vertices in the graph, and $d$ is the input feature depth), adjacency matrix $\boldsymbol{A} = (a_{uv}) \in \mathbb{R}^{n \times n}$ of the graph, and additional edge features $\mathbf{E} = (e_{uv}) \in \mathbb{R}^{n \times n \times l}$. Recall that a message passing layer of Gilmer et al. (2017) is of the form:

$$m_u^{t+1} = \sum_{v \in N(u)} M_t(h_u^t, h_v^t, e_{uv}) \tag{13a}$$

$$h_u^{t+1} = U_t(h_u^t, m_u^{t+1}) \tag{13b}$$

where $u, v$ are nodes in the graph, $h_u^t$ is the feature vector associated with $u$ in layer $t$, and $e_{uv}$ are additional edge features. We denote the number of output features of $M_t$ by $d'$.

In our setting we represent this data using a tensor $\mathbf{Y} \in \mathbb{R}^{n \times n \times (1+l+d)}$ where the first channel is the adjacency matrix $\boldsymbol{A}$, the next $l$ channels are edge features, and the last $d$ channels are diagonal matrices that hold $\boldsymbol{X}$.

Let us construct a message passing layer using our model:

1. Our first step is constructing an $n \times n \times (1 + l + 2d)$ tensor. In the first channels we put the adjacency matrix $\boldsymbol{A}$ and the edge features $\mathbf{E}$. In the next $d$ channels we replicate the features on the rows, and in the last $d$ channels we replicate features on the columns. The output tensor $\mathbf{Z}^1$ has the form $\mathbf{Z}_{u,v}^1 = [a_{uv}, e_{uv}, h_u^t, h_v^t]$.

2. Next, we copy the feature channels $[a_{uv}, e_{uv}, h_u^t]$ to the output tensor $\mathbf{Z}^2$. We then apply a multilayer perceptron (MLP) on the last $l + 2d$ feature dimensions of $\mathbf{Z}^1$ that approximates $M_t$ (Hornik, 1991). The output tensor of this stage is $\mathbf{Z}_{u,v}^2 = [a_{uv}, e_{uv}, h_u^t, M_t(h_u^t, h_v^t, e_{uv}) + \epsilon_1]$.

3. Next, we would like to perform point-wise multiplication $\mathbf{Z}_{u,v,1}^2 \odot \mathbf{Z}_{u,v,(l+d+2):end}^2$. This step would zero out the outputs of $M_t$ for non-adjacent nodes $u, v$. As this point-wise multiplication is not a part of our framework we can use an MLP on the feature dimension to approximate it and get $\mathbf{Z}_{u,v}^3 = [a_{uv}, e_{uv}, h_u^t, a_{uv}M_t(h_u^t, h_v^t) + \epsilon_2]$.

4. As before we copy the feature channels $[a_{uv}, e_{uv}, h_u^t]$. We now apply a sum over the rows ($v$ dimension) on the $M_t$ output channels. We put the output of this sum on the diagonal of $\mathbf{Z}^4$ in separate channels. We get $\mathbf{Z}_{u,v}^4 = [a_{uv}, e_{uv}, h_u^t, \delta_{uv} \sum_{w \in N(u)} M_t(h_u^t, h_w^t) + \epsilon_3]$, where $\delta_{uv}$ is the Kronecker delta. We get a tensor $\mathbf{Z}^4 \in \mathbb{R}^{n \times n \times (1+l+d+d')}$.

5. The last step is to apply an MLP to the last $d + d'$ feature channels of the diagonal of $\mathbf{Z}^4$. After this last step we have $\mathbf{Z}^5_{u,v} = [a_{uv}, e_{uv}, \delta_{uv}U_t(h^t_u, m^{t+1}_u) + \epsilon_4]$.

The errors $\epsilon_i$ depend on the approximation error of the MLP to the relevant function, the previous errors $\epsilon_{i-1}$ (for $i > 1$), and uniform bounds as-well as uniform continuity of the approximated functions. □

**Corollary 1.** *Our model can represent any message passing network to an arbitrary precision on compact sets. In other words, in terms of universality our model is at-least as powerful as* any *message passing neural network (MPNN) that falls into the framework of Gilmer et al. (2017).*

