# OpenReview forum: "Invariant and Equivariant Graph Networks"
_ICLR.cc/2019/Conference_

### Official Review · AnonReviewer1 · 2018-11-03
**Very interesting paper**

**Rating:** 9
**Confidence:** 4

**Review:**

This paper explores maximally expressive linear layers for jointly exchangeable data and in doing so presents a surprisingly expressive model. I have given it a strong accept because the paper takes a very well-studied area (convolutions on graphs) and manages to find a far more expressive model (in terms of numbers of parameters) than what was previously known by carefully exploring the implications of the equivariance assumptions implied by graph data. The result is particularly interesting because the same question was asked about exchangeable matrices (instead of *jointly* exchangeable matrices) by Hartford et al. [2018] which lead to a model with 4 bases instead of the 15 bases in this model, so the additional assumption of joint exchangeability (i.e. that any permutations applied to rows of a matrix must also be applied to columns - or equivalently, the indices of the rows and columns of a matrix refer to the same items / nodes) gives far more flexibility but without losing anything with respect to the Hartford et al result (because it can be recovered using a bipartite graph construction - described below). So we have a case where an additional assumption is both useful (in that it allows for the definition of a more flexible model) and benign (because it doesn't prevent the layer from being used on the data explored in Hartford et al.).

I only have a couple of concerns:
1 - I would have liked to see more discussion about why the two results differ to give readers intuition about where the extra flexibility comes from. The additional parameters of this paper come from having parameters associated with the diagonal (intuitively: self edges get treated differently to other edges) and having parameters for the transpose of the matrix (intuitively: incoming edges are different to outgoing edges). Neither of these assumptions apply in the exchangeable setting (where the matrix may not be square so the diagonal and transpose can't be used). Because these differences aren't explained, the synthetic tasks in the experimental section make this approach look artificially good in comparison to Hartford et al.  The tasks are explicitly designed to exploit these additional parameters - so framing the synthetic experiments as, "here are some simple functions for which we would need the additional parameters that we define" makes sense; but arguing that Hartford et al. "fail approximating rather simple functions" (page 7) is misleading because the functions are precisely the functions on which you would expect Hartford et al. to fail (because it's designed for a different setting).
2 - Those more familiar of the graph convolution literature will be more familiar with GCN [kipf et al. 2016] / GraphSAGE [Hamilton et al. 2017] / Monti et al [2017] / etc.. Most of these approaches are more restricted version of this work / Hartford et al. so we wouldn't expect them to perform any differently from the Hartford et al.  baseline on the synthetic dataset, but including them will strengthen the author's argument in favour of the work. I would have also liked to see a comparison to these methods in the the classification results.
3 - Appendix A - the 6 parameters for the symmetric case with zero diagonal reduces to the same 4 parameters from Hartford et al. if we constrained the diagonal to be zero in the output as well as the input. This is the case when you map an exchangeable matrix into a jointly exchangeable matrix by representing it as a bipartite graph [0, X; X^T, 0]. So the two results coincide for the exchangeable case. Might be worth pointing this out.

---

> ### Author Response · Authors · 2018-11-06
> **Addressing Reviewer 1 concerns**
>
> We thank the reviewer for the detailed review. Below we address the main concerns.
>
> --------------------------------------------------------------------------------------------------------------------------------
> Q:”so framing the synthetic experiments as, "here are some simple functions for which we would need the additional parameters that we define" makes sense; but arguing that Hartford et al. "fail approximating rather simple functions" (page 7) is misleading because the functions are precisely the functions on which you would expect Hartford et al. to fail”
>
> A: We agree with the reviewer and will change our wording accordingly.
> --------------------------------------------------------------------------------------------------------------------------------
> Q:”I would have liked to see more discussion about why the two results differ to give readers intuition about where the extra flexibility comes from”.  “the two results coincide for the exchangeable case”
>
> A: We agree with the reviewer that such a discussion will be helpful to the reader. We will add such a discussion (in addition to the short discussion at the end of Appendix 1).
> --------------------------------------------------------------------------------------------------------------------------------
> Q: Comparison to popular graph convolution methods (GCN [kipf et al. 2016] / GraphSAGE [Hamilton et al. 2017] / Monti et al [2017] / etc.).
>
> A: As discussed in our response to Reviewer 2, We will add a theoretical result that shows that our model is at least as powerful in terms of universality as [Kipf & Welling ICLR 2017].

---

### Official Review · AnonReviewer2 · 2018-11-05
**Nice combinatorics, but this is not what graph neural networks actually do**

**Rating:** 4
**Confidence:** 5

**Review:**

Given a graph G of n vertices, the activations at each level of a graph neural network (G-NN) for G
can be arranged in an n^k tensor T for some k. A fundamental criterion is that this tensor must be equivariant
to permutations of the vertices of G in the sense of each index of of T being permuted simultaneously.

This paper enumerates the set of all linear maps that satisfy this criterion, i.e., all linear maps
which (the authors claim) can serve as the analog of convolution in equivariant G-NNs.
The authors find that for invariant neural networks such maps span a space of dimension just b(k), whereas
for equivariant neural networks they span a space of dimension b(2k).

The proof of this result is simple, but elegant. It hinges on the fact that the set of tensor elements of
the same equality type is both closed and transitive under the permutation action. Therefore, the
dimensionality of the subspace in question is just the number of different identity types, i.e.,
partitions of either {1,...,k} or {1,...,2k}, depending on whether we are talking about invariance or
equivariance.

My problem with the paper is that the authors' model of G-NNs doesn't actually map to what is used
in practice or what is interesting and useful. Let me list my reservations in increasing order of significance.

1. The authors claim that they give a ``full characterization'' of equivariant layers. This is not true.
Equivariance means that there is *some* action of the symmetric group S_n on each layer, and wrt these actions
the network is equivariant. Collecting all the activations of a given layer together into a single object L,
this means that L is transformed according to some representation of S_n. Such a representation can always be
reduced into a direct sum of the irreducible representations of S_n. The authors only consider the case then
the representation is the k'th power of the permutation representation (technically called the defining
representation of the S_n). This corresponds to a specific choice of irreducibles and is not the most general case.
In fact, this is not an unnatural choice, and all G-NNs that I know follow this route.
Nonetheless, technically, saying that they consider all possible equivariant networks is not correct.

2. The paper does not discuss what happens when the input tensor is symmetric. On the surface this might seem
like a strength, since it just means that they can consider the more general case of undirected graphs (although
they should really say so). In reality, when considering higher order activations it is very misleading because
it leads to a massive overcounting of the dimensionality of the space of convolutions. In the case of k=2, for
example, the dimensionality for undirected graphs is probably closer to 5 than 15 for example (I didn't count).

3. Finally, and critically, in actual G-NNs, the aggregation operation in each layer is *not*
linear, in the sense that it involves a product of the activations of the previous layer with the adjacency
matrix (messages might be linear but they are only propagated along the edges of the graph).
In most cases this is motivated by making some reference to the geometric meaning of convolution,
the Weisfeiler-Lehman algorithm or message passing in graphical models. In any case, it is critical that the
graph topology be reintroduced into the network at each layer. The algebraic way to see it is that each layer
must mix the information from the vertices, edges, hyperedges, etc.. The model in this paper could only aggregated
edge information at the vertices. Vertex information could not be broadcast to neighboring vertices again.
The elemenary step of ``collecting vertex information from the neighbors but only the neighbors'' cannot be
realized in this model.

Therefore, I feel that the model used in this paper is rather uninteresting and irrelevant for practical
purposes. If the authors disagree, I would encourage them to explicitly write down how they think the model
can replicate one of the standard message passing networks. It is apparent from the 15 operations listed on
page 11 that they have nothing to do with the graph topology at all.

Minor gripes:

- I wouldn't call (3) and (4) fixed point equations, that's usually used in dynamical systems. Here there is
an entire subspace fixed by *all* permutations.

- Below (1), they probably mean that ``up to permutation vec(L)=vec(L^T)''.

---

> ### Author Response · Authors · 2018-11-06
> **Message passing using our method...**
>
> We thank the reviewer for the detailed review. The reviewer's main concern was the practicality of the method, and its inability to model message passing. We respectfully disagree. Below we show that our model can simulate standard message passing architectures in a simple way, as well as answer other concerns.
>
> --------------------------------------------------------------------------------------------------------------------------------
> Q: “Finally, and critically, in actual G-NNs, the aggregation operation in each layer is *not*
> Linear… I feel that the model used in this paper is rather uninteresting and irrelevant for practical purposes”
>
> A: We would like to address this from a few points of view.
> Message passing usually deals with vertex data and adjacency matrix. Our goal was to work with data such as general affinity matrices and higher order tensors. For example, pairwise distance matrix (order 2 tensor), or area/congruence of triplets (order 3 tensor) which are useful representation for geometric data, for instance.
> Also note that linear equivariant models like ours are already used in successful methods like PointNet [Qi et al. CVPR 2017], DeepSets [Zaheer et al. NIPS 2017], and [Hartford el al. ICML 2018].
> Having said that, our method is at-least as powerful in terms of universality as standard message passing (see next question).  We also note that our empirical results support the above mentioned theoretical results.
>
> --------------------------------------------------------------------------------------------------------------------------------
> Q: “I would encourage them to explicitly write down how they think the model
> can replicate one of the standard message passing networks”.
>
> A: Thank you for raising this question. Let us show:
>
> Proposition:
> Our model can represent Kipf & Welling’s message passing [Kipf & Welling ICLR 2017] to arbitrary precision.
>
> Proof:
> Consider input vertex data X\in R^{n x d} (n is the number of vertices in the graph, and d is the feature depth) and adjacency/affinity matrix A\in R^{n x n}  of the graph. In our setting we represent this data using a tensor Y\in R^{n x n x d+1} where the first channel is the adjacency matrix A and the last d channels are diagonal matrices that hold X. We would like to approximate the function Y \mapsto A*X. For simplicity we consider d=1 but the following generalizes readily to all d>1. A*X can be represented by first using our equivariant linear layer to replicate X values on the rows; denote this new matrix by Z \in R^{n x n x 2}, where the first channel of Z is A and the second is the replication of X. Now multiplying entrywise the two feature channels of Z followed by summing the rows (another equivariant 2->1 operator) will provide A*X.  Since pointwise product between features is not a part of our model we can approximate it to arbitrary precision using an MLP on the feature dimension that can be written as a series of linear equivariant operators and ReLUs. (Note that MLP on the feature dimension is the way PointNet and DeepSets work.) QED
>
> We will add this claim and proof to the paper.
>
> One immediate corollary of this proposition is that in terms of universality our model is at-least as powerful as Kipf & Welling message passing.
>
> --------------------------------------------------------------------------------------------------------------------------------
> Q: “The authors claim that they give a ``full characterization'' of equivariant layers”:
>
> A: We give a full characterization for equivariant maps for the natural action of the S_n on the graph tensors of all orders: consistent re-labeling of the graph nodes. The reviewer is correct in pointing out that not all irreducible representations are considered. We will be happy to rephrase our claim.
>
> --------------------------------------------------------------------------------------------------------------------------------
> Q: ”The paper does not discuss what happens when the input tensor is symmetric.”
>
> A: This question was addressed in Appendix 1. We add a relevant quote:
> “We note that in case the input matrix is symmetric, our basis reduces to 11 elements in the first layer. If we further assume the matrix has zero diagonal we get a 6 element basis in the first layer. In both cases our model is more expressive than the 4 element basis of Hartford et al. (2018) and as the output of the first layer (or other inner states) need not be symmetric nor have zero diagonal the deeper layers can potentially make good use of the full 15 element basis.”
> --------------------------------------------------------------------------------------------------------------------------------

---

> > ### Comment · AnonReviewer2 · 2018-11-12
> > **Response to rebuttal**
> >
> > Thanks for your response.
> >
> > I guess there is a distinction here that is blurred in the paper, but also, to some extent, in the literature. Learning from graphs and learning from subsets of {1,...,n} are not the same thing, even if both problems can be framed in terms of a hypergraph and both problems involve equivariance to the action of S_n.
> >
> > The graph of a molecular or a social network is much more than just a subset of V\times V: it has specific geometric structure and that's exactly what a G-NN is supposed to be able to latch onto. I was critical of the early Laplacian based graph G-NN papers exactly because they only dealt with the spectrum of the Laplacian, which is an easy way out: of course it is invariant to permutations, but it doesn't tell you much about the geometry.
> >
> > For this reason, being able to reproduce [Kip & Welling, 2017] doesn't impress me so much. Despite your claim, I don't regard that as a message passing algorithm, in fact, it looks like the word "message" doesn't even appear in the paper. [Gilmer et al, 2017] or any of the papers following it would be the bar if we are really talking about message passing.
> >
> > To refine my stance a little, I do understand that equivariance of tensors to the permutation action does come up in G-NNs. For example in [Kondor et al] at each vertex they take a higher order tensor and use contractions to reduce it to a number of lower order ones. Your results could be used to count how many different ways this can be done. But this is not exactly what you describe in the paper: the group is not S_n, but only a smaller symmetric group, etc.
> >
> > In summary, my main problem with the paper is that it overstates its scope. This paper is not really about graph neural networks, it is more about the "interactions between sets" nets. And it doesn't give a full characterization of equivariant layers, only a "more or less  full characterization" because the authors only consider the permutation action. The main result is Theorem 1, which is neat, but I wonder if it has the requisite technical depth or element of surprise to warrant a separate paper.

---

> > > ### Author Response · Authors · 2018-11-13
> > > **[Gilmer et al, 2017]**
> > >
> > > Thank you for taking the time to write a response and bringing up Gilmer's message passing formulation.
> > >
> > > 1) The Gilmer et al. 2017 bar: We have just uploaded a revised manuscript in which we prove that our model can approximate any message passing neural network of the general form introduced in [Gilmer et al. 2017].
> > >
> > > 2) “Interaction between sets” networks are not suitable for learning graphs: This claim is not entirely clear to us. The most popular way to represent a graph is by an affinity matrix that describes the interaction between every pair of nodes.  From our point of view, our work establishes a natural connection between “interaction between sets” networks and graph learning. Note that both approaches utilize the adjacency structure as well as node features, so the geometric structure of the graph is indeed visible and usable by our method.
> > >
> > > 3) Graph learning = message passing: Although message passing is a prominent graph learning method it is not the only approach to learning graph data. We introduce a method, based on a generalization of “interaction between sets” that theoretically contains the message passing framework. In any case, we believe seeking new/different methods to learn graphs is a worthy research goal.
> > >
> > > 4) Full characterization of linear layers: We have updated our contribution statement (in the introduction and abstract) to claim that we give a classification of *permutation* invariant/equivariant layers.

---

### Official Review · AnonReviewer3 · 2018-11-06
**Beautiful work -- problems with experiments**

**Rating:** 8
**Confidence:** 5

**Review:**

The paper presents a maximally expressive parameter-sharing scheme for hypergraphs, and in general when modeling the high order interactions between elements of a set. This setting is further generalized to multiple sets. The paper shows that the number of free parameters in invariant and equivariant layers corresponds to the different partitioning of the index-set of input and output tensors. Experimental results suggest that the proposed layer can outperform existing methods in supervised learning with graphs.

The paper presents a comprehensive generalization of a recently proposed model for interaction across sets, to the setting where some of these sets are identical. This is particularly useful and important due to its applications to graphs and hyper-graphs, as demonstrated in experiments.

Overall, I enjoyed reading the paper. My only concern is the experiments:

1) Some of the benchmark datasets for the proposed task as well as some well-known methods (see Battaglia et al’18 and references in there) are missing.

2) Applying the model of Hartford et al’18 to problems where interacting sets are identical is similar to applying convolution layer to a feature vector that is not equivariant to translation. (In both cases the equivariance group of data is a strict subgroup of the equivariance of the layer.)  Do you agree that for this reason, all the experiments on the synthetic dataset is flawed?

---

> ### Author Response · Authors · 2018-11-08
> **Addressing Reviewer 3 concerns**
>
> We thank the reviewer for the positive comments. Below we address the main concerns.
>
> Q: ”Applying the model of Hartford et al. to problems where interacting sets are identical is similar to applying convolution layer to a feature vector that is not equivariant to translation... Do you agree that for this reason, all the experiments on the synthetic dataset is flawed?”
>
> A: Our goal in performing the synthetic experiments was to quantify the expressive power that is  gained by adding our basis elements to [Hartford et al. 18]. We felt it is an informative experiment since [Hartford et al. 18] also discuss applying their model in the jointly exchangeable setting (page 3, second column, top paragraph).
> Having said that, we agree with the reviewer that [Hartford et al. 18] probably cannot handle such tasks by construction. As we mentioned in our response to Reviewer1 we will change the wording of this section to better reflect that this is *not* a failure of Hartford et al. but merely a setting outside their scope due to a different assumption on the symmetry group of the data.
> If the reviewers feel strongly about this experiment, we are open to replace it with a discussion.
>
> --------------------------------------------------------------------------------------------------------------------------------
> Q: “Some of the benchmark datasets for the proposed task as well as some well-known methods (see Battaglia et al’18 and references in there) are missing”.
>
> A: We did our best to survey and compare to the most related works on the dataset collection introduced in [Yanardag & Vishwanathan 2015]. These datasets contain graphs from multiple origins, where some of them consist of highly varying graph sizes (within the same dataset). In any case we will make the code available as soon as possible.
> --------------------------------------------------------------------------------------------------------------------------------

---

### Author Response · Authors · 2018-11-13
**Revision uploaded**

We thank all the reviewers for their time and effort. We have uploaded a revised manuscript in which we have incorporated the suggestions from the reviews and comments.

We want to highlight one specific addition to the manuscript (Appendix 3) which is a proof that our model can approximate any message passing neural network that falls in the framework of Gilmer et al. [2017] (i.e., the bar set by Reviewer2 after a fruitful discussion).

Note that these additions made us overflow by several lines which can be squeezed back if the reviewers require that.

---

### Public Comment · (anonymous) · 2018-11-19
**Missing related work in experiments.**

While the proposed solution was compared to a few algorithms, some recent state-of-the-art algorithms were omitted in the experiments sections, having a misleading impression on the performance of the author's algorithm. At least the following papers should be included and argued the differences with the author's approach.

[1] Ivanov et.al, Anonymous Walk Embeddings, ICML 2018
[2] Verma et.al, Hunt For The Unique, Stable, Sparse And Fast Feature Learning On Graphs, NIPS 2017

---

> ### Author Response · Authors · 2018-11-20
> **Response**
>
> Thank you for bringing to our attention these two recent works. We uploaded a revision.  These two works construct graph features that seem to be very useful for graph classification but are not directly related to our approach. We have added them to our table and updated the text accordingly. Indeed these methods outperform our method on most, but not all datasets.
> Note that we ran exactly the same 3-layer network on all datasets and still outperform other deep-learning based methods on the social network datasets.

---

### Meta-Review · Area_Chair1 · 2018-12-16
**Description of linear permutation invariant and equivariant layers**

**Confidence:** 4
**Recommendation:** Accept (Poster)

**Metareview:**

The paper provides a comprehensive study and generalisations of previous results on linear permutation invariant and equivariant operators / layers for the case of hypergraph data on multiple node sets. Reviewers indicate that the paper makes a particularly interesting and important contribution, with applications to graphs and hyper-graphs, as demonstrated in experiments.

A concern was raised that the paper could be overstating its scope. A point is that the model might not actually give a complete characterization, since the analysis considers permutation action only. The authors have rephrased the claim. Following comments of the reviewer, the authors have also revised the paper to include a discussion of how the model is capable of approximating message passing networks.

Two referees give the paper a strong support. One referee considers the paper ok, but not good enough. The authors have made convincing efforts to improve issues and address the concerns.